# iTryOn: Mastering Interactive Video Virtual Try-On with Spatial-Semantic Guidance

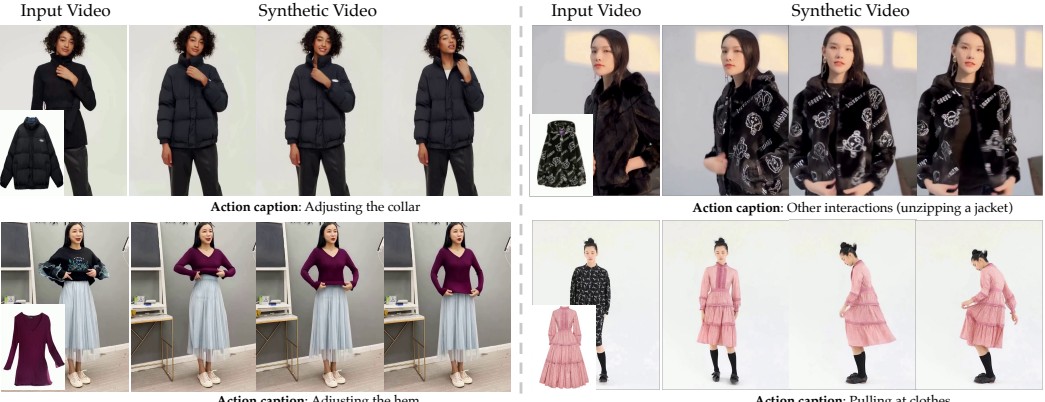

Figure 1: iTryOn synthesizes a diverse range of complex human-garment interactions guided by action captions. The examples showcase the model's ability to generate physically plausible deformations for various actions. (Best viewed in motion in the supplementary videos)

## Abstract

Video Virtual Try-On (VVT) aims to seamlessly replace a garment on a person in a video with a new one. While existing methods have made significant strides in maintaining temporal consistency, they are predominantly confined to non-interactive scenarios where models merely showcase garments. This limitation overlooks a crucial aspect of real-world apparel presentation: active human-garment interaction. To bridge this gap, we introduce and formalize a new challenging task: Interactive Video Virtual Try-On (Interactive VVT), where subjects in the video actively engage with their clothing (e.g., pulling a hem or unzipping a jacket). This task introduces unique challenges beyond simple texture preservation, including: (1) resolving the semantic ambiguity of interactions from standard pose information, and (2) learning complex garment deformations from video where interactive moments are sparse and brief. To address these challenges, we propose **iTryOn**, a novel framework built upon a large-scale video diffusion Transformer. iTryOn pioneers a multi-level interaction injection mechanism to guide the generation of complex dynamics. At the spatial level, we introduce a garment-agnostic 3D hand prior to provide fine-grained guidance for precise hand-garment contact, effectively resolving spatial ambiguity. At the semantic level, iTryOn leverages global captions for overall context and time-stamped action captions for localized interactions, synchronized via our novel Action-aware Rotational Position Embedding (A-RoPE). Furthermore, we design an action-aware constraint loss to stabilize training and focus the learning process on these critical interactive frames. To facilitate research and evaluation, we construct VVT-Interact, the first large-scale dataset for this task. Extensive experiments demonstrate that iTryOn not only achieves state-of-the-art performance on traditional VVT benchmarks but also establishes a commanding lead in the new interactive setting, marking a significant step towards more dynamic and controllable virtual try-on experiences.

# 1 INTRODUCTION

Generative models have achieved remarkable progress, catalyzing innovations across numerous domains, with virtual try-on emerging as a quintessential application in e-commerce and digital content creation. The field initially focused on image-based virtual try-on, where early methods leveraging Generative Adversarial Networks (GANs) (Xie et al., 2021a; He et al., 2022; Choi et al., 2021; Zhenyu et al., 2023; Xie et al., 2021b) have recently been surpassed by diffusion models (Kim et al., 2023; Xu et al., 2024a; Choi et al., 2024; Chong et al., 2024), which demonstrate superior fidelity in synthesizing realistic person-garment composites. However, static images fail to capture the dynamic interplay between a garment and human motion, a crucial factor for a comprehensive apparel assessment.

Consequently, research has shifted towards the more challenging yet practical task of Video Virtual Try-On (VVT). VVT aims to generate a temporally coherent video of a person wearing a new garment, capturing its drape, flow, and response to movement. A primary obstacle that distinguishes VVT from its image-based counterpart is ensuring spatiotemporal consistency—the seamless preservation of garment texture and structure across all video frames. A naive frame-by-frame application of image try-on methods invariably leads to flickering artifacts and temporal discontinuities. To overcome this, recent VVT methods (Xu et al., 2024b; Fang et al., 2024; Karras et al., 2024; Chong et al., 2025; Li et al., 2025b; Zuo et al., 2025) have successfully adapted powerful pre-trained diffusion models by incorporating temporal modules. These approaches leverage the strong priors learned from large-scale datasets to generate consistent and high-quality try-on videos, marking a significant advancement in the field. Despite this progress, existing VVT research shares a fundamental limitation: it operates exclusively within non-interactive scenarios. Current benchmarks and methods model a passive subject who simply moves or poses to display an outfit. However, the rise of live-streaming e-commerce has cultivated a new paradigm where presenters actively interact with their clothes, for example, stretching fabric to show elasticity or lifting a hem to reveal patterns. These interactions provide critical information to potential buyers but remain unaddressed by the VVT community. This discrepancy motivates us to define and tackle a new frontier: **Interactive Video Virtual Try-On (Interactive VVT)**.

The transition from non-interactive to interactive VVT introduces a unique set of challenges. The first is the semantic ambiguity of interactions. Standard conditioning signals like 2D keypoints (Yang et al., 2023) are insufficient as they lack 3D orientation and shape, making it impossible to distinguish an interactive gesture like tucking in a shirt from a non-interactive one. The second challenge is learning physical plausibility from sparse events. Interactive moments involving complex physics-driven deformations are often brief compared to simpler non-interactive segments. This imbalance creates a sparse and unstable supervisory signal, making it difficult for the model to converge on complex dynamics.

To overcome these hurdles, we propose iTryOn, a novel framework based on a large-scale video diffusion transformer that features two core innovations: a multi-level interaction injection mechanism and a targeted constraint loss. Our multi-level interaction injection mechanism resolves ambiguity by providing guidance at both spatial and semantic levels. At the spatial level, we introduce a garment-agnostic 3D hand prior to provide fine-grained guidance for the *how* of physical contact. This clean 3D reconstruction guides the model in generating accurate hand-garment contact, overcoming the limitations and information leakage of depth-based alternatives. At the semantic level, to address the *what* and *when* of an interaction, we introduce global captions for overall context and time-stamped action captions for localized control. To precisely synchronize these captions with their corresponding video segments, we design a novel Action-aware Rotational Position Embedding (A-RoPE). To address the challenge of learning from sparse events, we introduce an action-aware constraint loss. This loss function stabilizes the training process by strategically intensifying supervision on the critical but infrequent frames containing interactions. Finally, to support research and evaluation, we have curated VVT-Interact, the first large-scale dataset specifically for this task.

Our main contributions are summarized as follows: (1) We introduce and formalize a new task, Interactive Video Virtual Try-On (Interactive VVT), to address the limitations of existing methods in handling real-world human-garment interactions. To tackle this, we propose iTryOn, a novel framework based on a large-scale video diffusion transformer. (2) We propose a novel multi-level interaction injection mechanism that synergistically combines a garment-agnostic 3D hand prior for

spatial guidance with time-stamped action captions synchronized via A-RoPE for precise semantic control. (3) We design an action-aware constraint loss that amplifies supervision on interactive video segments, stabilizing training and improving the model's ability to learn complex garment dynamics from sparse events. (4) We construct VVT-Interact, the first dataset for interactive VVT research. Extensive experiments show that iTryOn not only excels in the new interactive setting but also achieves state-of-the-art performance on traditional VVT benchmarks.

## 2 RELATED WORK

### 2.1 VIDEO VIRTUAL TRY-ON

The recent proliferation of powerful open-source video generation models has catalyzed significant advancements in Video Virtual Try-On (VVT) (Xu et al., 2024b; Karras et al., 2024; Fang et al., 2024; Wang et al., 2024; Li et al., 2025a; Zheng et al., 2025; Chong et al., 2025; Li et al., 2025b; Zuo et al., 2025). Early diffusion-based methods focused on adapting image generation models for video tasks. For instance, ViViD (Fang et al., 2024) introduced a large-scale VVT dataset and repurposed an image diffusion model by inserting temporal motion modules to facilitate video-level synthesis. Subsequent works have increasingly leveraged the Diffusion Transformer (DiT) architecture, recognizing its superior capacity for spatiotemporal modeling. CatV$^2$TON (Chong et al., 2025) proposed a unified DiT-based framework for both image and video try-on. MagicTryOn (Li et al., 2025b) built upon the powerful Wan2.1 (Wan et al., 2025) backbone, enhancing garment fidelity by injecting fine-grained guidance in the form of detailed textual descriptions and contour line maps. More recently, DreamVVT (Zuo et al., 2025) introduced a two-stage pipeline, first generating keyframes with a multi-frame try-on model and then employing another powerful video generation model to synthesize the final video from these keyframes. While these methods excel at maintaining temporal consistency for passive motion, they universally neglect active human-garment interactions. This leaves the generation of complex physics-driven interaction dynamics as a major unaddressed problem. Our work pioneers the Interactive VVT task to fill this critical gap.

### 2.2 VIDEO GENERATION

Modern video generation is predominantly driven by diffusion models, with the Diffusion Transformer (DiT) architecture emerging as the state-of-the-art following the success of Sora (OpenAI, 2024). Early works like AnimateDiff (Guo et al., 2024) adapted image models with temporal modules, but recent top-performing models such as Hunyuan-DiT (Weijie Kong, 2024) and Wan2.1 (Wan et al., 2025) have embraced full spatiotemporal attention for superior cross-frame modeling. Our iTryOn framework builds upon this advanced lineage. We specifically adopt Wan2.1-VACE (Jiang et al., 2025) as our foundational backbone due to its strong controllable video generation capabilities. This allows us to frame video virtual try-on as a specialized video inpainting task, conditioned on a garment image for reference and human pose for structural control. Leveraging the powerful priors of Wan2.1-VACE significantly accelerates training convergence, enabling us to focus our efforts on the novel challenges of interactive video virtual try-on.

## 3 METHODOLOGY

### 3.1 PROBLEM FORMULATION

We formalize the task of Interactive Video Virtual Try-On (Interactive VVT). Given a source video $V_{\text{src}} \in \mathbb{R}^{T \times 3 \times H \times W}$ depicting a person interacting with their garment, and a target garment image $G \in \mathbb{R}^{3 \times H \times W}$, the objective is to synthesize a new video $\hat{V} \in \mathbb{R}^{T \times 3 \times H \times W}$. This output video must preserve the subject's identity and motion from $V_{\text{src}}$, while realistically rendering the target garment $G$ as it dynamically responds to the interaction. To achieve this, the task relies on a suite of conditional inputs $\mathcal{C}$, which includes the pose sequence $V_{\text{pose}}$, a clothing-agnostic representation $V_{\text{agn}}$, and specific guidance for the interaction itself. Therefore, the problem can be viewed as learning a mapping function $\mathcal{F}$ such that:

$$\hat{V} = \mathcal{F}(V_{\text{src}}, G, \mathcal{C}) \tag{1}$$

Successfully learning this mapping $\mathcal{F}$ is non-trivial and introduces several unique challenges not present in traditional VVT: (1) **Interaction Ambiguity**: Standard pose skeletons are ambiguous as their 2D projection collapses motion along the Z-axis, erasing crucial depth cues. For instance, the preparatory motion of a hand moving towards the chest to button a shirt becomes nearly invisible in 2D, depriving the model of the key "approaching" signal needed to anticipate contact and thus necessitating richer 3D guidance. (2) **Learning Physical Plausibility from Sparse Events**: While the ultimate goal is to generate physically plausible dynamics, learning this from video data presents a significant challenge. Interactive moments involving complex deformations are often brief and infrequent compared to simpler, non-interactive segments. This imbalance creates a sparse and unstable supervisory signal, where the gradient from easier, static frames can overwhelm the crucial but rare signal from interactive frames. Consequently, the model may fail to converge on complex dynamics, defaulting to simpler, non-interactive generations. (3) **Data Scarcity**: A significant bottleneck is the lack of suitable data. Existing VVT datasets consist almost entirely of non-interactive sequences and lack the fine-grained annotations required to supervise a model on specific interactions.

To address these challenges, we adopt a two-pronged approach. First, we construct a new large-scale dataset with detailed annotations designed to resolve ambiguity. Second, we propose the iTryOn framework, an architecture designed to generate physically plausible results based on this data.

### 3.2 DATA COLLECTION AND ANNOTATION OF VVT-INTERACT

#### 3.2.1 DATA SOURCING AND FILTERING

We initiated the process by extensively collecting video-garment pairs from e-commerce live streams and social media, which serve as rich sources for interactive clothing demonstrations. Recognizing the noisy nature of this raw data, we implemented a rigorous, multi-stage curation pipeline to ensure high quality and relevance. The pipeline first filters out unqualified data by: (1) removing pairs with low-resolution garment images; (2) discarding videos with low bitrates or significant visual artifacts; (3) excluding videos where the person occupies a small screen ratio; (4) eliminating instances where the garment is subject to unrecoverable occlusion; and (5) removing videos with scene cuts to ensure temporal continuity, using an automatic shot detection algorithm (Souček & Lokoč, 2020).

#### 3.2.2 VLM-BASED ANNOTATION FOR SEMANTIC GUIDANCE

The cornerstone of our dataset is its detailed annotation of interactions, designed to provide the multi-level semantic guidance required to resolve the interaction ambiguity challenge. We leveraged the advanced capabilities of Qwen-VL (Bai et al., 2025) to generate two distinct types of annotations: global captions and time-stamped action captions. Our annotation strategy proceeded as follows: (1) Global Caption Generation: We first prompted Qwen-VL to produce a high-level summary of the overall human motion in each video. This resulting global caption provides general context for the entire sequence. (2) Time-stamped Action Caption Generation: To pinpoint the exact temporal boundaries of interactions, we performed a fine-grained analysis. This involved tasking Qwen-VL to classify each frame as either "interactive" or "non-interactive," yielding a per-frame binary label. As the initial sequence of labels was often noisy, we applied morphological smoothing to denoise the predictions and identify continuous interaction segments. Finally, we combined these temporal boundaries with a pre-determined interaction category to automatically generate the time-stamped action captions, structured as ("action description", [start_frame, end_frame]).

The final VVT-Interact dataset consists of 5,292 high-quality video-garment pairs, covering six distinct interaction categories, each annotated with both a global caption and one or more time-stamped action captions. See Appendix A.3 for more details.

### 3.3 OVERVIEW OF THE ITRYON FRAMEWORK

The overall architecture of our proposed framework, iTryOn, is depicted in Figure 2. Built upon a conditional Diffusion Transformer (DiT) backbone, iTryOn is specifically designed to address the challenges outlined in our problem formulation. It processes a source video, a target garment, and a suite of conditional inputs to generate a realistic interactive try-on video. Guidance is injected into the DiT backbone through a set of parallel trainable modules. These include Context Blocks that process general body information (from pose and agnostic inputs) to ensure proper overall garment

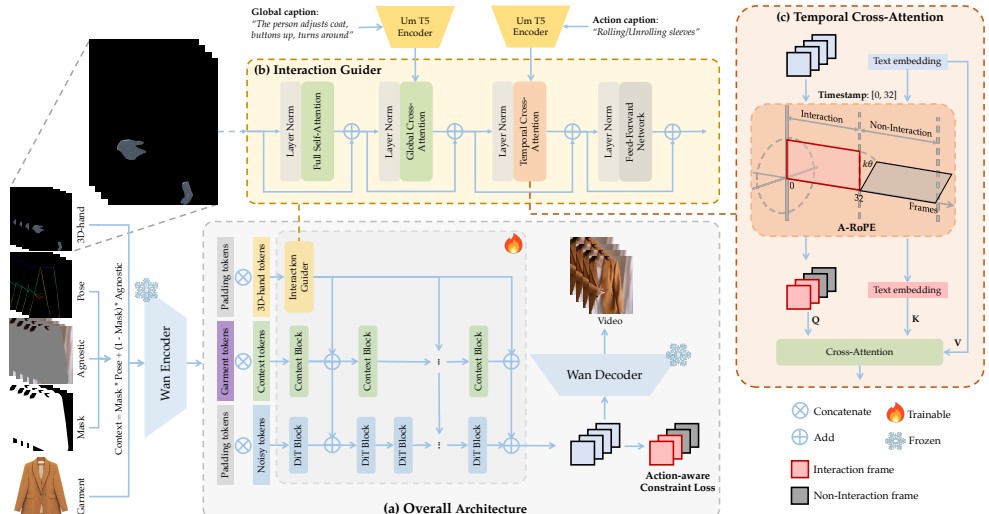

Figure 2: The iTryOn architecture. (a) A DiT backbone with parallel injection of general context and 3D-hand guidance from our Interaction Guider. An action-aware constraint loss focuses training on interaction frames. (b) The Interaction Guider module fuses spatial features with global and action-specific text prompts. (c) Our A-RoPE mechanism aligns action captions to their corresponding video segments via unique rotational position encodings in temporal cross-attention.

alignment, and our novel Interaction Guider which handles the fine-grained hand-garment contact. For efficiency, all guidance modules adopt a streamlined shared architecture, and we use only $\frac{N}{2}$ Context Blocks. The framework's core innovations are three-fold, each corresponding to a subsequent section: (1) A **fine-grained spatial guidance** mechanism processes 3D hand representations to control the precise physical contact in an interaction (Sec. 3.4). (2) An **action-aware semantic guidance** mechanism leverages time-stamped captions and our Action-aware Rotational Position Embedding (A-RoPE) to interpret the *what* and *when* of an interaction (Sec. 3.5). (3) An **action-aware constraint loss** is used during training to stabilize learning from sparse interactive events, focusing the model on complex dynamics to improve physical plausibility (Sec. 3.6).

The general data flow involves encoding all inputs into the latent space using a frozen Wan encoder, followed by an iterative denoising process within the DiT where our guidance is injected. The final denoised latents are then decoded back into the output video. The following sections will elaborate on each of these key components.

## 3.4 FINE-GRAINED SPATIAL GUIDANCE FOR HAND-GARMENT INTERACTION

Accurately modeling the *how* of an interaction requires resolving the spatial ambiguity inherent in 2D pose estimations (DWPose (Yang et al., 2023), DensePose (Güler et al., 2018)). This ambiguity is twofold: 2D projections lack hand shape, making it impossible to distinguish a pulling pinch from a pressing flat palm, and they lack hand orientation, failing to differentiate an interactive gesture from a non-interactive one. To address this fundamental limitation, we introduce a fine-grained spatial guidance mechanism. The choice of the geometric prior for this mechanism is critical. As illustrated in Figure 3, alternatives like hand depth are also flawed, suffering from information leakage that contaminates the conditioning signal.

In contrast, we select a 3D hand representation as our prior, which is both detailed and garment-agnostic. We leverage the HaMeR model (Pavlakos et al., 2024) to extract this 3D hand prior, denoted as $V_{\text{hand}} \in \mathcal{C}$. As depicted in Figure 2(a), this clean geometric signal is processed by a lightweight Interaction Guider module. Concurrently, broader contextual information from the pose $V_{\text{pose}}$ and agnostic video $V_{\text{agn}}$ is handled by parallel Context Blocks. The features from both the Interaction Guider and Context Blocks are then additively fused with the video tokens at each block of the DiT backbone. This injection of precise 3D hand geometry provides the model with explicit cues about hand shape, orientation, and proximity, guiding it to generate physically plausible and accurate hand-garment contact.

| Garment | Input Video | Parsing + Depth → Flawed Prior (Hand Depth) | Result w/ Flawed Prior | Our Prior (3D-hand) | Our Result |
| --- | --- | --- | --- | --- | --- |

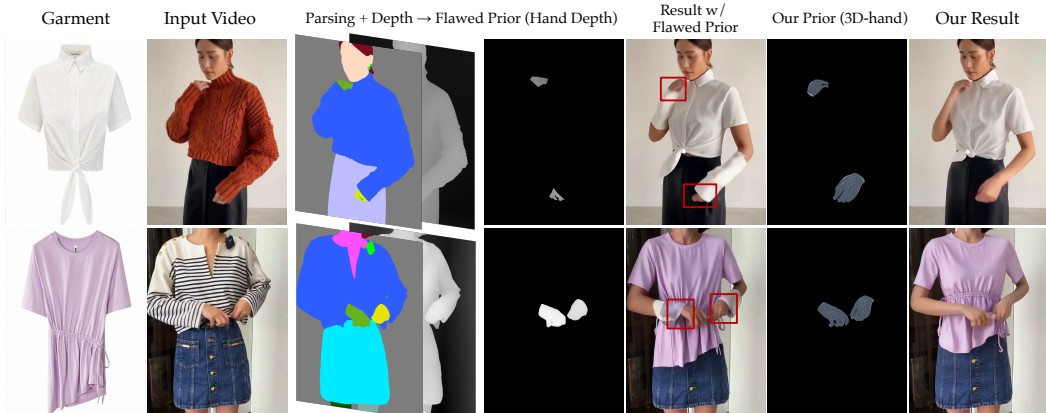

Figure 3: Visual justification for our garment-agnostic 3D hand prior. Deriving a "Hand Depth" prior from human parsing (Li et al., 2019) and video depth (Chen et al., 2025) suffers from critical information leakage. This flawed prior improperly retains source garment geometry, such as the sleeve cuff, leading directly to visible artifacts in the generated output. In contrast, our fully garment-agnostic 3D hand prior provides a clean signal, enabling the generation of plausible and artifact-free hand-garment contact.

### 3.5 ACTION-AWARE SEMANTIC GUIDANCE

While our spatial guidance resolves the *how* of an interaction, ambiguity remains concerning the *what* (the type of action) and the *when* (its precise timing). Although the global caption provides a high-level summary of the overall motion, we observed that its descriptions are often too generic to guide specific interactions (see **Appendix A.5.1** for detailed examples). This semantic ambiguity necessitates a more explicit form of guidance.

To address this, we introduce Action-aware Semantic Guidance, a mechanism composed of two key components: action captions for semantic specificity and an Action-aware Rotational Position Embedding (A-RoPE) for temporal precision. First, to specify the *what*, we complement the global caption with a categorical action caption drawn from a predefined set of interaction types. This provides the model with an unambiguous fine-grained signal about the intended action. However, interactions typically occur only within a short segment of the full video clip. Simply injecting this action caption via standard cross-attention can lead to temporal misalignment, where the semantic guidance "bleeds" into non-interactive frames. To enforce precise synchronization and control the *when*, we design A-RoPE, a novel embedding strategy inspired by MinT (Wu et al., 2025). As conceptualized in Figure 2(c), A-RoPE applies a scaled 1D-RoPE (Su et al., 2021) to distinguish between interactive and non-interactive segments based on their segment index $i$:

$$\hat{Q}_i = \text{A-RoPE}(Q_i, i) = \text{1D-RoPE}(Q_i, i \cdot k)$$
$$\hat{K}_i = \text{A-RoPE}(K_i, i) = \text{1D-RoPE}(K_i, i \cdot k)$$

(2)

where $k$ is a hyperparameter controlling the separation scale, which we set to 4 in our experiments. While A-RoPE is applied to the queries $Q_i$ of all video segments to maintain their temporal order, it is only applied to the keys $K_i$ derived from the action caption embeddings of the interactive segments. The value sequence $V$ is derived from the action caption embeddings without positional encoding. The final temporal cross-attention is computed as $\text{Attention}(\hat{Q}, \hat{K}, V)$. This design effectively creates a unique "temporal channel" for each interaction. By ensuring the positional encodings of a video segment's query $\hat{Q}_i$ and its corresponding action caption's key $\hat{K}_i$ are aligned, the attention mechanism is strongly biased to focus on the correct text-video pairing. This synchronizes the semantic guidance with high temporal fidelity, empowering the model to generate motions that are accurate in their specific timing.

### 3.6 ACTION-AWARE CONSTRAINT LOSS

To address the challenge of learning from sparse interactive events, we introduce an action-aware constraint loss (AC loss). Our guidance mechanisms provide the model with the necessary cues, but the inherent imbalance between frequent non-interactive frames and rare interactive frames can lead to training instability. The sparse gradient from complex deformations can be overwhelmed by the dense gradient from simpler frames, causing the model to neglect the crucial interaction dynamics. The AC loss counteracts this by amplifying the supervisory signal specifically on frames where interactions occur. The core idea is to strategically re-weight the standard diffusion loss, compelling the model to prioritize these critical moments. We leverage the temporal boundaries from our action captions to construct a binary mask $\mathbb{M}_{\text{action}}$ which is set to 1 for frames within an interaction segment and 0 otherwise. The overall training objective is formulated as:

$$\mathcal{L} = \mathbb{E}_{t,\mathbf{z}_t,c,v \sim \mathcal{N}(0,\mathbf{I})} \left[ \|v_\theta\left(\mathbf{z}_t, t, c\right) - v\|_2^2 \right] + \lambda \mathbb{E}_{t,\mathbf{z}_t,c,v \sim \mathcal{N}(0,\mathbf{I})} \left[ \|\mathbb{M}_{\text{action}} \odot \left(v_\theta\left(\mathbf{z}_t, t, c\right) - v\right)\|_2^2 \right],$$
(3)

where $z_t$ is the noisy latent at timestep $t$, $c$ represents the conditioning information, and $v_\theta(\cdot)$ is the v-prediction network. The first term is the standard diffusion loss computed over all frames. The second term weighted by a hyperparameter $\lambda$ (set to 0.5 in our experiments) applies an additional penalty exclusively to the latent features corresponding to the interaction frames, as selected by the element-wise multiplication with the mask $\mathbb{M}_{\text{action}}$. By applying this targeted supervisory signal, we prevent the model from ignoring the sparse but vital interaction dynamics. This focused training approach accelerates convergence on complex motions and significantly increases the success rate of generating the intended interaction, ultimately leading to more physically plausible results.

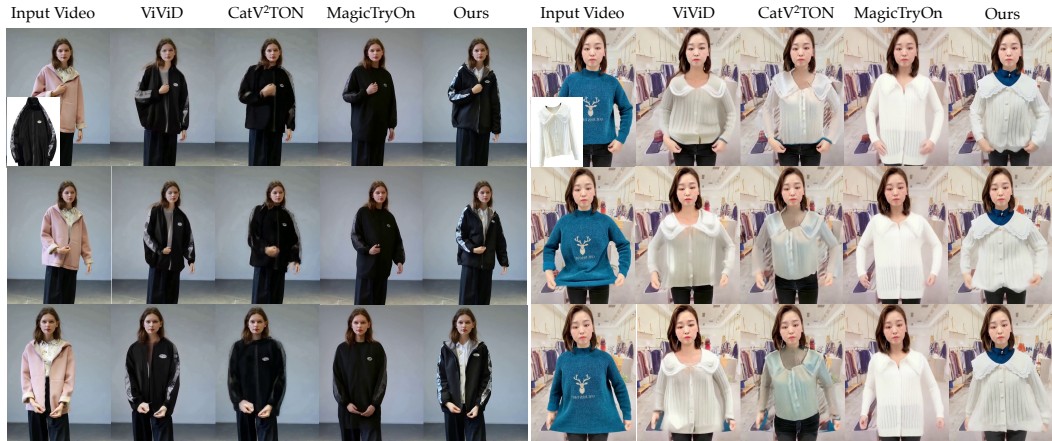

Figure 4: Qualitative comparison on the VVT-Interact dataset.

## 4 EXPERIMENTS

### 4.1 DATASETS AND METRICS

We conduct a comprehensive evaluation of our method on both traditional non-interactive and our newly proposed interactive video virtual try-on tasks. For the non-interactive VVT task, we benchmark our model on the widely-used ViViD dataset (Fang et al., 2024). The dataset comprises 7,759 paired videos for training and 180 videos for testing, all at a resolution of 624×832. To evaluate performance on our proposed interactive VVT task, we introduce the VVT-Interact dataset. Our dataset consists of 5,160 videos for training and 132 videos for testing. To ensure a fair and robust comparison against the non-interactive benchmark, the test set was curated to have a total of 10,692 frames, which is comparable to the 11,700 total test frames in the ViViD benchmark.

To assess the performance of our method, we employ a comprehensive set of established metrics. We use Structural Similarity (SSIM) (Wang et al., 2004) and Learned Perceptual Image Patch Similarity (LPIPS) (Zhang et al., 2018) to measure the similarity between generated frames and the ground truth. To evaluate the spatial quality and temporal consistency of the generated videos, we utilize Video Fréchet Inception Distance (VFID) (Dong et al., 2019) and Fréchet Video Distance

(FVD) (Unterthiner et al., 2018). We report VFID scores using two different feature extractors: I3D (VFID$_I$) (Carreira & Zisserman, 2017) and 3D-ResNeXt101 (VFID$_R$) (Hara et al., 2018).

## 4.2 IMPLEMENTATION DETAILS

Our model is initialized from the pre-trained Wan2.1-VACE (Jiang et al., 2025). We employ a two-stage training scheme. First, we finetune on the ViViD dataset for 10k steps with empty action captions. Then, we continue training on our VVT-Interact dataset for another 10k steps. Throughout all training, we use 81-frame video clips at 576×768 resolution with a per-GPU batch size of 1. We use the AdamW optimizer (Loshchilov & Hutter, 2017) with a learning rate of 1e-5. All experiments were conducted on 8 NVIDIA A100 (80GB) GPUs. For inference, we use 50 denoising steps and a CFG scale of 3.

Table 1: Quantitative comparison on the VVT-Interact dataset. The best and second-best results are denoted in **Bold** and with an underline. $p$ and $u$ denote the paired and unpaired settings, respectively.

| Method | VFID$_I^p \downarrow$ | VFID$_R^p \downarrow$ | FVD$^p \downarrow$ | SSIM↑ | LPIPS↓ | VFID$_I^u \downarrow$ | VFID$_R^u \downarrow$ | FVD$^u \downarrow$ |
|---|---|---|---|---|---|---|---|---|
| ViViD (Fang et al., 2024) | 29.8272 | 1.2735 | 468.4750 | 0.7259 | 0.1637 | 36.5179 | 1.6128 | 482.2153 |
| CatV$^2$TON (Chong et al., 2025) | 26.9919 | 2.2692 | 533.2168 | 0.7761 | 0.1434 | 36.4519 | 2.6764 | 542.4718 |
| MagicTryOn (Li et al., 2025b) | 27.6716 | 2.6022 | 431.7865 | 0.7649 | 0.1702 | 36.0322 | 3.3669 | 432.3735 |
| iTryOn (ours) | **22.4640** | **0.6033** | **380.5578** | **0.7849** | **0.1217** | **35.0479** | **1.2378** | **393.0552** |

Table 2: Quantitative comparison on the ViViD dataset.

| Method | Params | VFID$_I^p \downarrow$ | VFID$_R^p \downarrow$ | SSIM↑ | LPIPS↓ | VFID$_I^u \downarrow$ | VFID$_R^u \downarrow$ |
|---|---|---|---|---|---|---|---|
| ViViD (Fang et al., 2024) | 2B | 17.2924 | 0.6209 | 0.8029 | 0.1221 | 21.8032 | 0.8212 |
| CatV$^2$TON (Chong et al., 2025) | 5B | 13.5962 | 0.2963 | 0.8727 | 0.0639 | 19.5131 | 0.5283 |
| MagicTryOn (Li et al., 2025b) | 14B | 12.1988 | 0.2346 | 0.8841 | 0.0815 | 17.5710 | 0.5073 |
| DreamVVT (Zuo et al., 2025) | - | 11.0180 | 0.2549 | 0.8737 | **0.0619** | 16.9468 | 0.4285 |
| iTryOn (ours) | 2B | **8.4322** | **0.0876** | **0.8944** | 0.0679 | **13.2806** | **0.2293** |

## 4.3 QUANTITATIVE RESULTS

We quantitatively evaluate iTryOn against state-of-the-art methods on both interactive and non-interactive benchmarks with results presented in Table 1 and Table 2, respectively. As shown in Table 1, iTryOn establishes a commanding lead on our VVT-Interact dataset, significantly outperforming all baseline methods across every metric. This substantial margin validates the effectiveness of our proposed multi-level interaction injection mechanism in handling complex dynamics that prior methods neglect. Beyond its primary task, iTryOn also demonstrates superior performance and high efficiency on the ViViD benchmark (Table 2). Our 2B parameter model achieves state-of-the-art results while being up to 7x more parameter-efficient than competitors like MagicTryOn (14B). This confirms that our framework is not only a novel solution for interactive VVT but also a powerful and highly efficient model for the traditional task. A detailed analysis is provided in Appendix A.6.

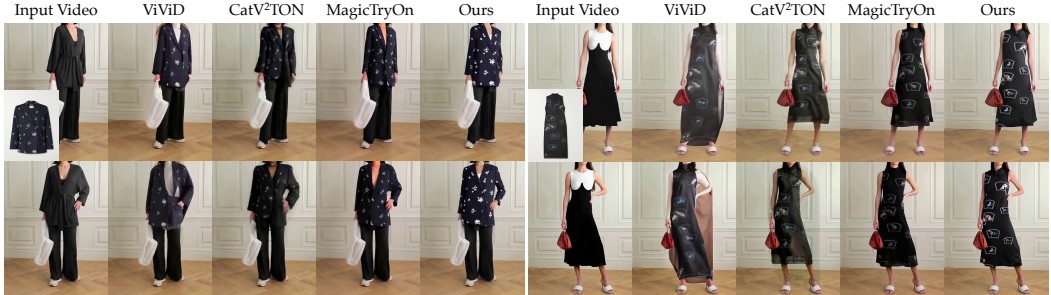

Figure 5: Qualitative comparison on the ViViD dataset.

## 4.4 QUALITATIVE RESULTS

We provide qualitative comparisons in Figure 4 to visually substantiate our quantitative dominance in the interactive setting. Figure 4 illustrates the failure of existing methods on our VVT-Interact

dataset. When faced with a zippering motion, baseline approaches either generate physically implausible deformations (ViViD) or completely misinterpret the action, producing a simple hand-gliding motion (CatV$^2$TON, MagicTryOn). Similarly, for a hem-pulling action, they render a static unresponsive garment. In contrast, iTryOn is the only method that successfully synthesizes these interactions with high physical realism, accurately depicting the fabric zippering and stretching in response to user actions. These results powerfully demonstrate the unique capability of our framework to interpret and render complex dynamic interactions. Furthermore, as shown in Figure 6.(1), iTryOn also excels in the traditional non-interactive setting, consistently preserving garment details and intricate patterns more faithfully than competing methods.

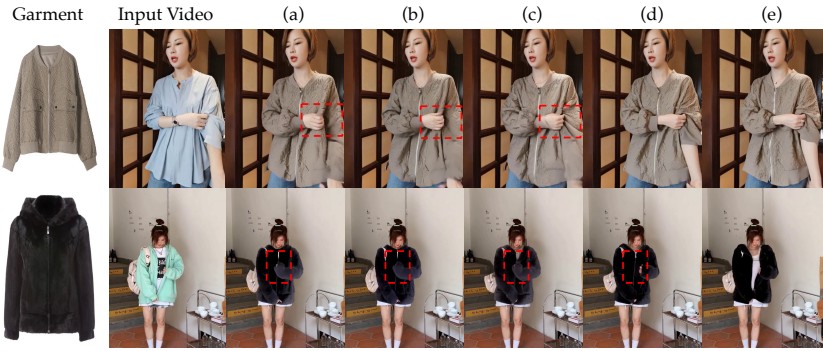

Figure 6: Visual comparison of different variants on the VVT-Interact dataset.

Table 3: Ablation study of each component on the VVT-Interact dataset.

| ID | Method | $\text{VFID}^p_I \downarrow$ | $\text{VFID}^p_R \downarrow$ | $\text{FVD}^p \downarrow$ | SSIM↑ | LPIPS↓ | $\text{VFID}^u_I \downarrow$ | $\text{VFID}^u_R \downarrow$ | $\text{FVD}^u \downarrow$ |
|---|---|---|---|---|---|---|---|---|---|
| (a) | Baseline | 27.1203 | 1.3242 | 419.0978 | 0.7720 | 0.1670 | 36.8623 | 1.5241 | 414.3908 |
| (b) | (a) + Data | 26.6500 | 0.8028 | 394.1350 | 0.7759 | 0.1338 | 36.5853 | 1.3677 | 405.4380 |
| (c) | (b) + Spatial Guidance | 24.8549 | 0.8284 | 384.7537 | 0.7833 | 0.1291 | 35.6026 | 1.4318 | 397.2745 |
| (d) | (c) + Semantic Guidance | 22.7558 | 0.6652 | **379.7348** | 0.7848 | 0.1228 | 35.4903 | 1.2442 | **391.8533** |
| (e) | (d) + AC loss (ours) | **22.4640** | **0.6033** | 380.5578 | **0.7849** | **0.1217** | **35.0479** | 1.2378 | 393.0552 |

## 4.5 ABLATION STUDIES

Our ablation study summarized in Table 3 and Figure 6.(2), systematically validates that our performance stems from our novel architecture, not merely from additional training data. Critically, the results demonstrate that simply training on our VVT-Interact dataset (b) is insufficient for the interactive task. While metrics show a slight improvement over the baseline, (b) visually confirms that the model still fails to synthesize any meaningful interactions. This underscores that existing VVT architectures cannot learn complex dynamics from data alone. Furthermore, while adding Spatial Guidance (c) enables physical hand-garment contact, it cannot resolve the inherent semantic ambiguity. The model knows where the hands are but not what they are doing. This ambiguity is effectively addressed by our Semantic Guidance (d), which provides the necessary intent. With the AC loss (e) providing further refinement, the study confirms that it is the synergistic combination of our proposed spatial and semantic guidance mechanisms that is essential for achieving high-fidelity interactive video virtual try-on.

## 5 CONCLUSION

In this work, we introduced and formalized the new task of Interactive Video Virtual Try-On (Interactive VVT). To facilitate research in this new domain, we proposed a novel framework iTryOn, and constructed the first corresponding dataset VVT-Interact. To generate physically plausible and controllable interactions, iTryOn incorporates a multi-level interaction injection mechanism for both spatial and semantic guidance, and an action-aware constraint loss to enhance supervision on key interactive frames. Extensive experiments demonstrate that iTryOn not only significantly outperforms existing methods on our VVT-Interact benchmark but also achieves state-of-the-art performance on the traditional non-interactive ViViD dataset. We believe our work marks a significant step towards more dynamic, realistic, and controllable virtual try-on applications.

## 6 ETHICS STATEMENT

We have carefully considered the ethical implications of our work, particularly concerning the creation of the VVT-Interact dataset and the application of our generative model. The dataset was constructed using publicly available videos from trusted sources where content creators have implicitly or explicitly consented to the public sharing of their content. To further protect personal identity, our data processing pipeline and model design are inherently privacy-preserving. The virtual try-on task is formulated to retain the head and other identifying features of the subject from the source video. The model's generative process is strictly confined to inpainting the garment and relevant limbs (hands, arms, feet), and does not reconstruct or generate facial features. This design choice mitigates the potential for misuse in creating deepfakes or otherwise compromising personal identity. We are committed to the responsible development of generative technologies in alignment with the ICLR Code of Ethics.

## 7 REPRODUCIBILITY STATEMENT

To ensure the reproducibility of our findings and foster future research in the new domain of Interactive VVT, we commit to the public release of our VVT-Interact dataset. A detailed description of the iTryOn architecture, including its novel guidance mechanisms and loss function, is provided in Section 3. The appendix further details our implementation, hyperparameter settings, and the data collection and annotation protocols. The dataset is currently undergoing final curation to ensure it is clean and easy to use, and will be released upon the publication of this work.

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

# A APPENDIX

## A.1 LIMITATIONS AND FUTURE WORK

While iTryOn marks a significant advancement in virtual try-on, we identify several key areas for future exploration and improvement.

**Quantitative Metrics for Interaction Fidelity.** A primary challenge in the nascent field of Interactive VVT is the lack of specialized evaluation metrics. While we employ standard pixel-level (SSIM, LPIPS) and video-level (FVD, VFID) metrics, they primarily assess overall visual quality and temporal consistency rather than the specific correctness of a physical interaction. For instance, these metrics cannot distinguish between a physically plausible fabric stretch and a visually coherent but incorrect one. A crucial direction for future work is therefore the development of novel metrics designed to explicitly quantify the fidelity of human-garment interactions, potentially by analyzing fine-grained physical dynamics or semantic correctness.

**Data Distribution.** As detailed in Sec. A.3, the distribution of interaction categories in our VVT-Interact dataset is imbalanced. This can lead to variations in the model's generative quality across different action types, with more frequent categories potentially being learned more effectively. Consequently, a key direction for future work is the targeted collection and augmentation of data for underrepresented categories. We believe that a more balanced dataset will enhance the model's generalization capabilities and ensure consistently high-quality results across the full spectrum of human-garment interactions.

**Model Scale.** The current implementation of iTryOn is based on a 2B parameter model, a choice constrained by available computational resources. It is well-established in the literature that larger foundation models often encapsulate more robust priors, including an implicit understanding of physical principles and richer world knowledge. Therefore, a promising avenue for future research involves scaling up the iTryOn framework by integrating it with a larger pre-trained video diffusion backbone. We anticipate that leveraging a more powerful foundation model would further improve the physical plausibility and overall fidelity of the generated interactions.

## A.2 THE USE OF LARGE LANGUAGE MODELS

In the preparation of this manuscript, large language models (LLMs) were utilized as a writing aid, specifically for grammar correction and sentence polishing to improve clarity and readability. The core research, including the formulation of ideas, experimental design, data analysis, and interpretation of results, was conducted entirely by the authors. LLMs played no role in the scientific contributions of this work.

## A.3 DATA ANNOTATION PIPELINE

This section provides a detailed description of the annotation pipeline used to create the VVT-Interact dataset, supplementing the overview provided in Sec. 3.2.2 of the main paper. Our pipeline consists of two primary components: VLM-based annotation for semantic guidance and 3D hand prior generation.

### A.3.1 VLM-BASED ANNOTATION FOR SEMANTIC GUIDANCE

We utilized the Qwen-VL-32B model (Bai et al., 2025) for all semantic annotations due to its superior performance in our preliminary evaluations. The process was divided into caption generation and timestamp annotation.

**Caption and Interaction Type Annotation**. To generate both the global caption and the categorical action caption efficiently, we designed a single-pass inference prompt. This prompt instructs the VLM to produce a JSON object containing a high-level motion description and the specific interaction type. The six predefined interaction categories and their respective proportions in the final dataset are: Adjusting the collar (41.50%), Adjusting the hem (32.56%), Rolling/Unrolling sleeves (9.92%), Putting on/Taking off clothes (7.97%), Pulling at clothes (5.06%), and Other interactions (2.98%).

The complete prompt is provided below:

> Your task is to analyze the elements in the images and answer the following true/false and open-ended questions. For true/false questions, your answer must be only true/false. For open-ended questions, the word count of the answer must be controlled within 65 words. The format for each question is "Title": Question. Answer each question sequentially in JSON format, using the "Title" as the key for each item. Answer the following questions referencing the images True/False Questions "interaction_available": Is the person in the video interacting with the clothing? "valid_interaction": Does the person's interaction with the clothing belong to one of the following types: [Putting on/Taking off clothes, Rolling/Unrolling sleeves, Adjusting the collar, Adjusting the hem, Pulling at clothes, Other interactions]? Open-ended Questions "interaction_type": What type of interaction is the person performing with the clothing? Please select from the options above. If not, please answer "N/A". "describe": These are sequential frames extracted from a video (one frame per second). Please describe the key actions or motion shown across these frames. Only return the overall motion description without any additional text. (answer in English)

**Timestamp Annotation and Smoothing**. To acquire precise temporal boundaries for interactions, we tasked Qwen-VL-32B with a per-frame binary classification task. The raw binary labels from the VLM, however, often contain noise (e.g., isolated misclassifications). To address this, we treat the sequence of labels as a 1D signal and apply morphological operations (specifically, morphological opening followed by closing). This procedure effectively removes spurious predictions and forms coherent, continuous interaction segments, from which we extract the start and end timestamps. The detailed prompt is as follows:

> Analyze the image to determine if the person is performing a manipulative interaction with their clothing. We are only interested in purposeful actions that change or adjust the garment. Your task is to distinguish between active manipulation, passive contact, and no contact. A 'manipulative interaction' is defined as any action intended to adjust, fasten, or change the state of the garment. Consider the action 'true' only if it meets the criteria below: Pulling, tugging, or stretching the fabric to adjust its fit or position. Zipping or unzipping. Buttoning or unbuttoning. Rolling up or down sleeves. Adjusting a collar, lapel, cuff, or hemline. Putting on or taking off the garment. Actively smoothing out a wrinkle or crease with pressure. Consider the action 'false' in all other cases, especially the following: No Contact: Any pose where the hands do not touch the clothing (e.g., arms crossed, hands at sides, gesturing in the air). Passive Contact: Gently stroking or caressing the surface of the fabric without intent to adjust it. Resting Contact: Simply resting a hand or arm on the clothing without applying force to move or change it. Incidental Contact: Posing with a hand in a pocket, where the primary action isn't adjusting the pocket itself. Based on these detailed definitions, is a manipulative interaction occurring in the image? Please respond with only the word 'true' or 'false'.

### A.3.2 VLM MODEL SELECTION

To select the optimal VLM for our annotation pipeline, we conducted a comparative study between the Qwen-VL and Gemma3 series, chosen for their strong performance and efficiency. We manually annotated a test set of 1,000 frames for the binary interaction classification task and evaluated each model's performance. The results are summarized in Table 4. As shown, Qwen-VL-32B achieves

Table 4: Comparison of different VLMs for the per-frame interaction annotation task. Qwen-VL-32B demonstrates the best overall performance, particularly in F1-score and precision.

| Model | Accuracy | Precision | Recall | F1-score |
|---|---|---|---|---|
| Qwen-VL 7B | 62.8 | 60.0 | 75.0 | 66.6 |
| **Qwen-VL 32B** | **74.1** | **79.7** | 69.8 | **74.4** |
| Gemma3 12B | 57.3 | 54.9 | 77.6 | 64.3 |
| Gemma3 27B | 56.9 | 54.0 | **87.2** | 66.7 |

the highest F1-score and precision. We note that this annotation task has inherent ambiguity, particularly in identifying the exact start and end frames of an interaction. Given this context, the performance of Qwen-VL-32B is considered highly effective for our large-scale automated annotation requirements.

### A.4 3D HAND PRIOR ANNOTATION

The 3D hand prior is generated using HaMeR (Pavlakos et al., 2024), which we applied on a per-frame basis to estimate the 3D hand mesh and pose from the input video. The resulting 3D information was then rendered into 2D image representations to be used as spatial guidance. A manual inspection of a random subset of the data revealed a high accuracy rate, exceeding 95%. Furthermore, the overall framework is robust to minor inaccuracies in the 3D hand prior, as the DWpose (Yang et al., 2023) features provide a foundational and reliable representation of the overall body and hand position.

### A.5 FURTHER DETAILS ON ACTION-AWARE SEMANTIC GUIDANCE

This section provides a deeper analysis of our Action-aware Semantic Guidance module. We first present qualitative examples to motivate the need for explicit action captions, and then provide detailed quantitative ablation studies to validate the effectiveness of each component.

**Example 1: Rolling Sleeves**          **Example 2: Adjusting the Hem**

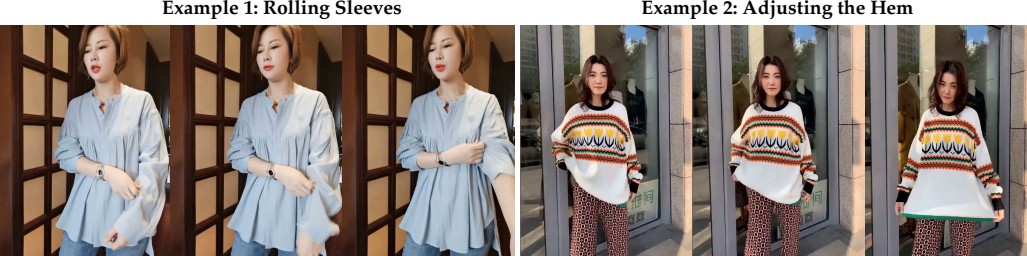

**Global caption**: The person stands still, **adjusts** shirt sleeves and collar, moves hands, and shifts body slightly.
**Our Action caption**: Rolling/Unrolling sleeves

**Global caption**: The person stands still, turns slightly, and **adjusts** the sweater.
**Our Action caption**: Adjusting the hem

Figure 7: Visual motivation for our Action-aware Semantic Guidance. These examples from our VVT-Interact dataset highlight the semantic ambiguity of VLM-generated global captions. Although the ground-truth interactions are distinct (rolling sleeves vs. adjusting the hem), both are imprecisely described with the generic verb "adjusts". Our categorical action captions resolve this ambiguity, providing the model with a clear and actionable signal required for high-fidelity interaction synthesis.

#### A.5.1 MOTIVATION: AMBIGUITY IN GLOBAL CAPTIONS

As stated in the main paper, a key motivation for our work is the inherent ambiguity of high-level motion descriptions generated by VLMs. While these global captions provide a useful summary, they often fail to capture the specific nature of a human-garment interaction, using generic verbs for distinct actions. This semantic ambiguity acts as a confusing supervisory signal, causing the model to default to the easier task of generating a non-interactive try-on rather than attempting a specific complex interaction. Figure 7 presents concrete examples from our VVT-Interact dataset that illustrate this problem.

#### A.5.2 QUANTITATIVE ABLATION STUDY

To validate the contribution of our proposed components, we conduct a detailed ablation study. As discussed in the main paper, the transition from model (c) to (d) in Table 3 highlights the impact of our full Semantic Guidance module. We dissect this gain by incrementally adding the action caption and A-RoPE to the baseline with spatial guidance (c). The results are presented in Table 5.

- Benefit of Action Captions: Comparing model (c) and (d'), we observe a consistent improvement across most metrics after introducing the time-stamped action captions. This confirms that providing the model with an explicit semantic signal about the action's type is crucial for resolving the ambiguity demonstrated above and improving generation quality.

- **Crucial Role of A-RoPE:** The subsequent addition of A-RoPE in model (d) yields another performance leap. The improvement is particularly pronounced in metrics sensitive to temporal consistency, such as FVD. This validates our hypothesis that precisely synchronizing the textual guidance with the corresponding video frames is critical. A-RoPE prevents the semantic information from "bleeding" into non-interactive frames and empowers the model to generate actions with accurate timing.

Table 5: Detailed ablation study on the components of Semantic Guidance.

| ID | Method | $\text{VFID}_I^p \downarrow$ | $\text{VFID}_R^p \downarrow$ | $\text{FVD}^p \downarrow$ | SSIM↑ | LPIPS↓ | $\text{VFID}_I^u \downarrow$ | $\text{VFID}_R^u \downarrow$ | $\text{FVD}^u \downarrow$ |
|---|---|---|---|---|---|---|---|---|---|
| (c) | Baseline + Data + Spatial Guidance | 24.8549 | 0.8284 | 384.7537 | 0.7833 | 0.1291 | 35.6026 | 1.4318 | 397.2745 |
| (d') | (c) + Action caption | 23.5618 | **0.5584** | **378.4681** | 0.7823 | 0.1232 | 36.0608 | 1.2831 | 393.4641 |
| (d) | (c) + Action caption + A-RoPE | **22.7558** | 0.6652 | 379.7348 | **0.7848** | **0.1228** | **35.4903** | **1.2442** | **391.8533** |

A key hyperparameter in our A-RoPE design is the separation scale $k$ from Equation 2. This parameter controls how distinctly different action segments are encoded in the positional space. To determine the optimal value, we performed an ablation study on $k$, with results shown in Table 6.

Table 6: Ablation study on the separation scale hyperparameter $k$ in A-RoPE. The value $k = 4$ yields the best overall performance.

| Method | $\text{VFID}_I^p \downarrow$ | $\text{VFID}_R^p \downarrow$ | $\text{FVD}^p \downarrow$ | SSIM↑ | LPIPS↓ | $\text{VFID}_I^u \downarrow$ | $\text{VFID}_R^u \downarrow$ | $\text{FVD}^u \downarrow$ |
|---|---|---|---|---|---|---|---|---|
| $k=2$ | 23.8789 | **0.6124** | 380.3563 | 0.7825 | 0.1247 | 35.6738 | 1.7365 | 392.2614 |
| $k=4$ | **22.7558** | 0.6652 | **379.7348** | **0.7848** | **0.1228** | **35.4903** | **1.2442** | **391.8533** |
| $k=6$ | 22.8378 | 0.6852 | 379.9243 | 0.7844 | 0.1341 | 35.5785 | 1.2652 | 392.8633 |

In conclusion, this detailed analysis validates our approach to Action-aware Semantic Guidance. We have first demonstrated that categorical action captions are essential for resolving the critical semantic ambiguity found in global prompts, which can cause the model to default to simpler non-interactive generations. Subsequently, we have shown that our proposed A-RoPE mechanism is crucial for enforcing the temporal precision required to synchronize this powerful guidance. The synergistic combination of these two components is key to empowering the model to generate accurate interactions.

## A.6 ANALYSIS OF STATE-OF-THE-ART PERFORMANCE ON NON-INTERACTIVE VVT

Readers may note that iTryOn, despite its novel components being specifically designed for interactive scenarios, achieves state-of-the-art performance on the non-interactive ViViD benchmark (Table 2). This section clarifies that this strong performance is not an anomaly but the result of two key factors: our strategic choice of a foundational backbone and the application of advanced general-purpose training and inference strategies.

Table 7: Ablation of general-purpose enhancements on the ViViD dataset.

| ID | Method | $\text{VFID}_I^p \downarrow$ | $\text{VFID}_R^p \downarrow$ | SSIM↑ | LPIPS↓ | $\text{VFID}_I^u \downarrow$ | $\text{VFID}_R^u \downarrow$ |
|---|---|---|---|---|---|---|---|
| (1) | Wan2.1-VACE (SFT) (Jiang et al., 2025) | 9.6956 | 0.1367 | 0.8892 | 0.0704 | 14.1810 | 0.4060 |
| (2) | (1) + Loss Weight | 9.3765 | 0.1308 | 0.8894 | 0.0703 | 13.6870 | **0.1890** |
| (3) | (2) + Interval Guidance (Kynkäänniemi et al., 2024) | **8.4697** | **0.0831** | **0.8948** | **0.0663** | **13.3776** | 0.1911 |

## A.6.1 A FOUNDATIONAL BACKBONE INHERENTLY SUITED FOR VVT

While contemporary methods like MagicTryOn (Li et al., 2025b) and DreamVVT (Zuo et al., 2025) are also built on powerful video generation models, our choice of Wan2.1-VACE (Jiang et al., 2025) as a backbone offers a distinct advantage for the VVT task. Wan2.1-VACE is a versatile controllable video synthesis model pre-trained for tasks like reference-guided editing. This allows us to frame VVT as a task perfectly aligned with the backbone's pre-trained capabilities: a specialized video inpainting/editing task where the garment image serves as the high-fidelity reference and the human

pose provides structural control. The model has already learned strong priors for preserving the textural identity of a reference object across complex video motions. Consequently, our framework inherits a superior capacity for maintaining garment fidelity and temporal coherence, forming a powerful baseline even on non-interactive tasks.

### A.6.2 ADVANCED TRAINING AND INFERENCE STRATEGIES

Beyond the strong backbone, our performance is further boosted by general-purpose techniques that enhance training efficiency and generation quality. To demonstrate their impact, we conduct a detailed ablation study, starting from our Wan2.1-VACE backbone fine-tuned on the ViViD dataset and incrementally adding each strategy. The results are presented in Table 7. Our methodology incorporates two key enhancements:

- **Loss Weighting:** During training, we apply a specific loss weighting scheme[1] common in flow matching, which helps accelerate convergence and stabilize the fine-tuning process. As shown by the improvement from model (1) to (2) in Table 7, this simple addition yields a consistent performance gain across all metrics.
- **Interval Guidance:** During inference, we employ Interval Guidance (Kynkäänniemi et al., 2024) for Classifier-Free Guidance (CFG). Instead of applying CFG throughout the entire denoising process, which can lead to oversaturation and artifacts, Interval Guidance restricts its application to a specific range of sampling steps (e.g., the first 10%-40%). The transition from model (2) to (3) demonstrates the substantial benefit of this technique.

This quantitative analysis confirms that a significant portion of our SOTA performance on ViViD stems from these powerful, general-purpose strategies. When combined with our inherently suitable backbone, they create a highly effective framework for traditional VVT, independent of our interaction-specific innovations.

---

[1]https://github.com/modelscope/DiffSynth-Studio/blob/main/diffsynth/schedulers/flow_match.py

