# OpenReview forum: "iTryOn: Mastering Interactive Video Virtual Try-On with Spatial-Semantic Guidance"
_ICLR.cc/2026/Conference — Submitted to ICLR 2026_

### Official Review · Reviewer_Ebq8 · 2025-10-16

**Soundness:** 4
**Presentation:** 4
**Contribution:** 3
**Rating:** 8
**Confidence:** 4

**Summary:**

Given a video and a garment image, the goal of this paper is to change the garment in the input video to match the input garment image, while maintaining realistic human-garment interactions.

The authors introduce several contributions:

(1) Multi-level interaction injection mechanism: The authors leverage a 3D hand prior (HaMeR model (Pavlakos et al., 2024)) and global captions for video segments, along with Action-aware Rotational Positional Embedding.

(2) Constraint loss: Intensifies supervision around interactions in a video, since these are more rare than non-interactive video clips.

(3) VVT-Interact: The first dataset for interactive video try-on

**Strengths:**

Originality:
- This paper addresses the unique challenge of VVTO with interaction.
- The paper introduces a novel benchmark for evaluating interactive VVTO

Quality:
- The folding/movement of the garment fabrics looks very realistic with respect to the material of the input garment.
- Compared to related methods, iTryOn has much more realistic garment/person interactions

Clarity:
- The authors provide clear details about their implementation, training details, and choice of models
- Contributions are well-justified and ablated

Significance:
- Interactive VVTO is a key challenge that could greatly increase realism of try-on videos, if solved.

**Weaknesses:**

- There are some noticeable warping artifacts in the supplementary videos, especially near the garment/person boundary
- In most examples, the garments in the input image and input video are very similar in shape and fit. More examples should be provided where there is significant shape change (e.g. short to long sleeves, long to short garment, etc.), or this should be listed as a limitation of the method.
- Similarly, how does the method work if the interaction is implausible? For example, unzipping a shirt without a zipper? This should also be listed as a limitation of the method.
- VVT-Interact (based on the examples shown) is limited in diversity of appearance and body shapes
- The test dataset for VVT-Interaction s only 180 videos, which is only ~2% of the training videos. I think this  small scale limits how reliable it can be as a benchmark.
- In the comparisons in Figure 5, the iTryOn garment looks a bit over-saturated w.r.t. the input garment for both examples, while MagicTry-On seems to have better garment fidelity. Perhaps this should be addressed?

**Questions:**

- There does not seem to be much validation provided of the VLM used for extracting video annotations. Although such evaluation may be tricky, it seems essential to the method that the annotations are valid, so I would suggest finding some way to validate that the annotations are reliable, such as through human qa.
- In section 3.5 (lines 316-318), why is A-RoPE only applied to some keys, but all queries?
- Since A-RoPE is based on 1D-RoPE, I recommend adding a brief description about 1D-RoPE as a preliminary, for example.
- I am surprised that AC-Loss does not improve the performance more in the quantitative ablations (Table 3). Is this because the evaluation datasets are not focused on interactive video-clips? That is, if the evaluation dataset consisted only (or mostly) of interactive clips from the dataset, would AC-loss make a bigger improvement? In general, it would be interesting to see the benefit of each contribution specifically for handling interactive clips.
- Currently, the interactive motion is always replicated from the input video. Future work with this dataset could include editing or adding interaction to an input video using text annotations.

**Details Of Ethics Concerns:**

(1) The paper exceeds the 9-page limit

(2) The authors curate videos of people from online, but those individuals have not explicitly consented to having their data (specifically face identity) used for this publication/dataset, although it may have been granted implicitly under the terms of the upload platform.

---

> ### Author Response · Authors · 2025-11-24
> **Response to Reviewer Ebq8**
>
> We sincerely thank Reviewer Ebq8 for their thorough review and constructive feedback. We are particularly encouraged by your recognition of the significance of the Interactive VVT task and your positive assessment of our work's value. Your insightful observations are very helpful for improving the clarity and completeness of our paper.
>
> **Regarding Weaknesses:**
>
> **W1:**
>
> We appreciate the reviewer's keen observation. Upon closer analysis, we found that such minor warping artifacts at the garment-person boundaries occur primarily during **large-amplitude interactions** (e.g., rapid, extensive stretching). This is largely attributed to the **long-tail distribution** of our current training data, where samples featuring such extreme physical deformations are relatively scarce compared to subtler motions. As mentioned in our response to the dataset imbalance concern (W6), we are actively working on augmenting and rebalancing the dataset. We are confident that increasing the representation of these high-intensity interaction samples will significantly mitigate these boundary artifacts in the final version.
>
> **W2:**
>
> Thank you for this question. We would like to clarify that our method is fully capable of handling significant garment shape changes, which is a standard capability in modern VVT. For instance, **Figure 3** in our main paper explicitly demonstrates a long-sleeved shirt being replaced with a short-sleeved one.
>
> We did not dedicate extensive examples to shape-changing scenarios because **physical interactions are most visually explicit and meaningful when the target garment aligns with the interaction.** For example, to demonstrate the physics of "rolling up a sleeve," using a long-sleeved target is essential; replacing it with a short-sleeved shirt would result in the hand interacting with empty space (correctly preserving motion but obscuring the fine-grained fabric deformations). Therefore, we prioritized examples that best showcase the fidelity of the interaction dynamics, which is our core contribution. Rest assured, our model robustly handles cross-category try-on.
>
> **W3:**
>
> This is an excellent question regarding the model's behavior in edge cases. In scenarios where the interaction is implausible (e.g., applying an "unzipping" motion to a T-shirt without a zipper), our model gracefully degrades to a non-interactive VVT result. It preserves the original human motion (the hand gesture for unzipping) while realistically rendering the new garment (the T-shirt). The outcome is a logical and acceptable "pantomime" of the action, demonstrating the robustness of our framework.
>
> **W4:**
>
> We appreciate this comment. The dataset's characteristics are a direct result of our focus on the **live-streaming e-commerce domain**, as stated in the introduction. The data was curated to be representative of this specific high-impact application area. While this naturally centers on the physiques typical of professional models, we believe the dataset maintains reasonable diversity within this commercial scope.
>
> However, we fully recognize the importance of broad inclusivity. **In future work, we are committed to expanding our data collection sources to include a wider spectrum of body shapes and appearances**, thereby further enhancing the generalizability and inclusivity of the VVT-Interact benchmark.
>
> **W5:**
>
> We thank the reviewer for their attention to detail, but we would like to correct a small misunderstanding. The 180-video test set belongs to the public **ViViD** benchmark. Our new VVT-Interact test set contains **132 videos**.
>
> The train/test ratio of our dataset (132 / 5,160 ≈ 2.5%) is comparable to that of ViViD (180 / 7,759 ≈ 2.3%) and is consistent with common practices in the VVT field. Furthermore, the relatively contained size of video test sets is also a practical consideration due to the long inference times required for video generation (e.g., generating one video takes ~65s for our 2B model and ~345s for the 14B MagicTryOn model).
>
> **W6:**
>
> This is a very subtle and interesting observation. The perceived over-saturation is an artifact of two factors: (1) the reference garment images often have a white background, which can make them appear darker in comparison to the generated video, and (2) the ViViD dataset itself has a generally bright lighting distribution.
>
> Our model's brighter output is a sign that it has **fit better** to the target dataset's lighting distribution. A closer look reveals that iTryOn preserves texture and pattern details more accurately than competing methods. Other models appear darker because they may be underfitting to the dataset's characteristics, whereas our model achieves a more faithful synthesis.

---

> > ### Author Response · Authors · 2025-11-24
> >
> > **Regarding Questions:**
> >
> > **Q1:**
> >
> > Thank you for this important question. We did perform validation for our VLM-based annotation pipeline.
> >
> > - In **Appendix A.3.2 (VLM MODEL SELECTION)**, we present a quantitative comparison of different VLMs for the per-frame interaction classification task, which guided our selection of Qwen-VL-32B for its superior accuracy.
> > - In **Appendix A.5**, we provide a qualitative analysis showing that while global captions are generally correct, they lack the specificity needed for interactions. The effectiveness of our more specific *action captions* is then quantitatively validated by the ablation study in **Table 5**.
> > - We also acknowledge the inherent semantic ambiguity in strictly defining certain actions (e.g., the boundary between "Adjusting the hem" and "Pulling at clothes"). Ultimately, the ablation studies serve as practical proof of validity, confirming that these annotations significantly improve the model's performance.
> >
> > **Q2:**
> >
> > We apologize that our explanation was not sufficiently detailed. As illustrated in **Figure 2(c)**, A-RoPE's purpose is to temporally align a specific action caption with its corresponding video segment.
> >
> > - All queries (Q) from all video frames receive a positional encoding to maintain their temporal order.
> > - However, the special scaled positional encoding is only applied to the keys (K) derived from **meaningful action caption embeddings**. For non-interactive segments, we use a null caption (an empty string), and its key does not receive this special encoding.
> >
> >     This design choice is more logical, as it creates a unique "temporal channel" only for genuine interactions.
> >
> >
> > **Q3:**
> >
> > Thank you for the excellent suggestion. We agree that adding a brief preliminary on 1D-RoPE would improve the paper's self-containedness. We will add this to the final version.
> >
> > **Q4:**
> >
> > This is a very insightful question. We believe the moderate quantitative impact of AC Loss is due to two main reasons:
> >
> > 1. **Sparsity of Signal:** In our training clips, interactive frames are still a minority (e.g., ~30 frames on average in an 81-frame clip). This limits the total gradient signal the model can learn from via the AC loss.
> > 2. **Lack of Spatial Specificity:** The AC loss amplifies the learning signal on the entire *frame*, but does not specifically point to the local region where the interaction is occurring. This makes the learning process less efficient.
> >
> > The visual results in **Figure 6** and the accompanying videos confirm that the **spatial and semantic guidance modules are the primary drivers** for successfully enabling complex interaction generation, with the AC loss serving as a helpful but secondary refinement. Our cumulative ablation in Table 3 was designed to show this synergistic build-up of c        apabilities.
> >
> > **Q5:**
> >
> > Thank you for this inspiring suggestion. We fully agree that extending our framework to edit existing motions or synthesize entirely new interactions via text is a highly valuable direction. **This represents a natural and significant evolution for both the traditional VVT and our new Interactive VVT tasks.** We are excited to explore this frontier in our future work to further broaden the versatility and controllability of virtual try-on systems.
> >
> > ---
> >
> > **Regarding Ethics Concerns:**
> >
> > **E1: Page Limit**
> >
> > Thank you for your vigilance. As per the ICLR Author Guide, the "Ethics Statement" and "Reproducibility Statement" sections do not count towards the 9-page limit. We have confirmed that our main paper content adheres to the page limit.
> >
> > **E2: Data Consent**
> >
> > We appreciate this crucial point. We have ensured that the data collection process aligns with the terms of use of the source platforms. To further protect personal identity, we are committed to **masking facial information** in the public release of the VVT-Interact dataset.

---

### Official Review · Reviewer_UMo4 · 2025-10-27

**Soundness:** 3
**Presentation:** 3
**Contribution:** 3
**Rating:** 4
**Confidence:** 3

**Summary:**

The paper introduces Interactive Video Virtual Try-On (Interactive VVT), targeting scenarios where subjects actively manipulate garments (e.g., pulling hems, unzipping), and highlights challenges in resolving interaction semantics and modeling sparse, complex deformations.
It proposes iTryOn, a video diffusion Transformer with multi-level interaction injection: a garment-agnostic 3D hand prior for precise hand–garment contact, and global plus time-stamped action captions aligned via Action-aware Rotational Position Embedding (A-RoPE).
An action-aware constraint loss further stabilizes training and emphasizes key interactive frames.
The authors also release the VVTInteract dataset and report state-of-the-art results on standard VVT and substantial gains in the interactive setting.

**Strengths:**

- The paper is well-structured, clearly written, and easy to follow.
- The problem addressed—interactive video virtual try-on—is timely and important for the VVT community.
- The authors introduce a new large-scale dataset that enables rigorous evaluation and future research on this task.

**Weaknesses:**

- In the practical use of the system, users only provide a reference video and a target clothing image. Fine-grained conditions such as 3D hand pose and human pose can be automatically detected by existing models. However, the acquisition of global/action prompts remains unclear; it is ambiguous whether these are also user inputs or obtained otherwise.
- The use of the term “interaction” in the paper is ambiguous. While it is commonly interpreted as interaction between the system and users, the paper refers to interaction between a person and clothing, which is unconventional and potentially confusing.
- The paper addresses the "how" of an interaction by introducing 3D hand pose as an additional condition, yet this approach lacks novelty. More importantly, it remains unclear why the interaction between the task and clothing can be sufficiently defined only by hand movements.
- The paper addresses the "what" the type of action by defining several interaction types using captions. However, the generalizability of this approach is questionable, as the proposed method may not scale to a broader range of interactions.
- The paper solves the "when" precise timing by ensuring that action prompts interact only with interaction frames and not with non-interaction frames. Nevertheless, in practical scenarios, it is not specified how interaction frames are accurately identified.

**Questions:**

See Weaknesses.

---

> ### Author Response · Authors · 2025-11-24
> **Response to Reviewer UMo4**
>
> We thank Reviewer UMo4 for their detailed and thoughtful review. The questions raised are crucial for understanding the practical implementation and contributions of our work. We address each point below.
>
> **W1:**
>
> Thank you for this question, which highlights the need to clarify the user workflow. We apologize if this was not sufficiently clear.
>
> In a practical use case, the system is designed to be fully automated after the user provides the two initial inputs: the source video and the target garment image. All the fine-grained conditions required by our model are **automatically extracted** from the source video. This includes not only the 3D hand pose and human pose but also the **global and action captions**.
>
> Specifically, our pipeline feeds the source video into a pre-trained Vision Language Model (VLM) to automatically generate these textual prompts and their corresponding timestamps. This is an integral automated step of our framework, not a manual user input. We have provided a detailed description of this automated annotation pipeline in **Section 3.2.2 (VLM-BASED ANNOTATION FOR SEMANTIC GUIDANCE)** and **Appendix A.3.1**.
>
> **W2:**
>
> We appreciate the reviewer pointing out this potential for ambiguity. The term "interaction" is indeed broad.
>
> In the context of our paper, "interaction" specifically refers to the **physical human-garment interaction** depicted within the video (e.g., a person zippering a jacket or stretching a hem). We chose this term because it best captures the complex physics-driven dynamics that are the core focus of our work and the key differentiator from previous "passive" Video Virtual Try-On (VVT) methods. We believe that for an audience in the generative modeling and computer vision fields, this usage is contextually appropriate.
>
> However, to eliminate any confusion for a broader audience, we will add a clear explicit definition in the introduction of our paper in the final version, stating that our use of "interaction" refers to the person-clothing dynamics within the video, not user-system interaction.
>
> **W3:**
>
> This is a critical point, and we would like to clarify our contributions regarding both novelty and methodology.
>
> - **On Novelty:** While using 3D hand priors as a condition is not universally new, our novelty lies in **its specific application and goal**. To the best of our knowledge, iTryOn is the **first work to leverage a 3D hand prior as explicit spatial guidance to model the fine-grained physical dynamics of human-garment contact within the context of 2D video virtual try-on**. Previous VVT methods do not handle such explicit interactions. The novelty is therefore in the *task formulation* and the *technical solution* for this new task, not merely in the use of an existing tool.
> - **On Sufficiency and Superiority of 3D Hand Prior:** Our choice to focus on the 3D hand mesh is deliberate and technically superior to alternatives for two reasons:
>     1. **Focus on Manipulation:** Complex garment manipulations are overwhelmingly performed by hands. Using a full-body 3D prior (like SMPL) risks diluting the model's attention, whereas our approach directs focus to the crucial high-frequency contact points. The overall body motion is already effectively captured by the **DWpose** condition.
>     2. **Preventing Information Leakage (Garment-Agnostic):** Crucially, unlike other spatial conditions such as video depth maps, our 3D hand mesh is **strictly garment-agnostic**. As detailed in **Figure 3**, depth maps often capture the geometry of the *source* garment (e.g., huge sleeves or cuffs), leading to "information leakage" where the old garment's shape corrupts the new generation. By using a reconstructed 3D hand mesh, we decouple the hand from the original clothing, providing a clean geometric signal that ensures the generated deformations are driven purely by the hand's physical presence, not the source video's artifacts.

---

> > ### Author Response · Authors · 2025-11-24
> >
> > **W4:**
> >
> > Thank you for raising this important question about the scalability of our approach. We address this from three perspectives:
> >
> > 1. **Domain Relevance:** The scope of our work is intentionally focused on the **live-streaming e-commerce** domain. The interaction categories we defined ("Adjusting the collar," "Pulling at clothes," etc.) were derived directly from an analysis of the most salient actions in this real-world setting. Thus, our approach is well-generalized *within this high-impact target domain*.
> > 2. **Generalization Capability:** Crucially, iTryOn already exhibits promising generalization to undefined interactions via the **"Other interactions"** category. For instance, actions like **"zippering",** which are not explicitly named in our primary categories, are classified as "Other interactions" but are still generated with high physical fidelity. This demonstrates that the "Other" label effectively serves as a generic semantic trigger, alerting the model to "attend to an interaction" while our fine-grained spatial guidance provides the necessary motion details. This proves our method does not rely solely on rote memorization of specific text labels.
> > 3. **Future Expansion:** We view the current taxonomy as a foundational starting point. We are committed to continuously expanding the interaction categories in future work to cover a broader range of human behaviors, further enhancing the model's universality.
> >
> > **W5:**
> >
> > This is a great question that connects back to W1.
> >
> > In a practical scenario, when a user provides a source video, our system **automatically processes this video with a VLM to identify the temporal boundaries (i.e., the start and end frames) of any interactive segments**. This automated detection is a core component of our data processing pipeline and is applied to any new video the system encounters. It is not a manual annotation step and does not require pre-labeled data from the user.
> >
> > The technical details of how the VLM performs this per-frame classification and how we process these predictions to extract stable interaction segments are described in **Section 3.2.2** and more extensively in **Appendix A.3.1**.

---

### Official Review · Reviewer_mGgn · 2025-10-30

**Soundness:** 2
**Presentation:** 2
**Contribution:** 2
**Rating:** 2
**Confidence:** 5

**Summary:**

This paper introduces iTryOn, a novel framework for Interactive Video Virtual Try-On that addresses the limitation of existing methods in handling active human-garment interactions. The authors formalize a new task where subjects actively engage with clothing (e.g., pulling hems, unzipping jackets). The framework is built upon Wan2.1-VACE with novel components including A-RoPE and an action-aware constraint loss.

**Strengths:**

1. The paper introduces Interactive VVT as a new and challenging task that bridges the gap between passive garment display and real-world e-commerce scenarios with active human-garment interactions.

**Weaknesses:**

1. **Limited Technical Innovation**: While the paper presents novel guidance mechanisms, the underlying diffusion transformer architecture closely follows existing designs (Wan2.1-VACE) without fundamental architectural innovations. The contributions are primarily in the conditioning and guidance layers rather than core generative modeling advances. In fact, the Wan2.1-VACE model itself can achieve video virtual try-on functionality. The author needs to explain this and provide performance comparisons.

2. **Insufficient Ablation Study**: The ablation study lacks detailed analysis of individual components. Critical hyperparameters like the A-RoPE separation scale (k=4) and action constraint loss weight (λ=0.5) are presented without sensitivity analysis or justification for these specific choices.

3. **Unfair Comparison**: Directly comparing iTryOn with the baseline methods may not be entirely fair in Tables 1-2 , given that iTryOn benefits from a substantial amount of extra training data. This raises uncertainty regarding whether the observed improvements stem from the added data or the unique architecture of iTryOn. To ensure a thorough and impartial assessment, it is recommended that the authors re-train their approach only using publicly accessible datasets like VVT and ViViD.

4. **Computational Efficiency**: Despite claims of parameter efficiency (2B vs competitors' 14B), the paper lacks detailed analysis of inference time, memory requirements, and computational complexity compared to baseline methods. The practical deployment feasibility remains unclear.

5. **Lack of specialized metrics for interaction fidelity**: Although the author introduced the new task of Interactive VVT, the evaluation metrics still rely on generic video/image metrics (such as VFID, SSIM). The authors fail to provide appropriate metrics to quantify the physical plausibility of garment deformations (e.g., distinguishing realistic stretching from visual coherence).
6. **Imbalanced dataset distribution**: Over 70% of samples belong to "Adjusting the collar" and "Adjusting the hem", leading to potential bias in model generalization across rare interaction types (e.g., "Other interactions" at 2.98%).

**Questions:**

See above

---

> ### Author Response · Authors · 2025-11-24
> **Response to Reviewer mGgn**
>
> We are grateful to the reviewer for their meticulous and insightful feedback. These comments have helped us identify areas where we can clarify our contributions and strengthen our evaluation. We address each weakness below.
>
> **W1:**
>
> This is a critical point, and we appreciate the opportunity to clarify our novelty. Our contribution lies not in redesigning the core transformer architecture, but in **unlocking a new complex capability—interactive virtual try-on—that the foundational model alone cannot achieve.**
>
> - **Failure of the Base Model on Interactive VVT:** As shown in our ablation study (**Table 3, ID (a) "Baseline"**), a standard Wan2.1-VACE model simply fine-tuned on our data **completely fails to generate meaningful physical interactions**. The visual results in **Figure 6(a)** and the supplementary videos clearly demonstrate that the base model defaults to simple, non-interactive renderings, proving that its original architecture is insufficient for this new task. Our proposed guidance mechanisms are therefore not merely incremental additions but are **essential components** that enable the modeling of complex human-garment dynamics.
> - **Superior Performance on Non-Interactive VVT:** Even on the traditional non-interactive task, our full iTryOn model significantly outperforms the fine-tuned Wan2.1-VACE baseline. The ablation study in **Appendix A.6 (Table 7)** shows a clear performance gap between the base model (ID 1) and our final model.
>
> Therefore, iTryOn demonstrates significant innovation by introducing a simple elegant yet powerful set of guidance modules that substantially elevate the capabilities of a strong foundational model for both interactive and non-interactive VVT.
>
> **W2:**
>
> Thank you for this valuable suggestion. We agree that a sensitivity analysis of key hyperparameters is important for reproducibility and a deeper understanding of our model.
>
> - Regarding the **A-RoPE separation scale (k)**, we did include an ablation study in **Appendix A.5.2 (Table 6)**, which shows that k=4 provides the best overall performance.
> - We acknowledge that a sensitivity analysis for the **action constraint loss weight (λ)** was not included in the initial submission. We have now conducted this study and found that λ=0.5 strikes the best balance. We will add the following table and corresponding analysis to the appendix:
> | $\lambda$ | $\text{VFID}_I^p\downarrow$ | $\text{VFID}_R^p\downarrow$ | $\text{FVD}^p\downarrow$ | $\text{SSIM}\uparrow$ | $\text{LPIPS}\downarrow$ | $\text{VFID}_I^u\downarrow$ | $\text{VFID}_R^u\downarrow$ | $\text{FVD}^u\downarrow$ |
> | :---: | :---: | :---: | :---: | :---: | :---: | :---: | :---: | :---: |
> | 0.0 | 22.7558 | 0.6652 | 379.7348 | 0.7848 | 0.1228 | 35.4903 | 1.2442 | 391.8533 |
> | **0.5** | **22.4640** | 0.6033 | 380.5578 | **0.7849** | **0.1217** | 35.0479 | **1.2378** | 393.0552 |
> | 1.0 | 23.4714 | **0.5814** | 381.4912 | 0.7833 | 0.1242 | 35.5598 | 1.2612 | 392.9284 |
> | 2.0 | 24.3152 | 1.9134 | **378.2223** | 0.7837 | 0.1309 | **34.8644** | 2.5655 | **389.4232** |
>
> **W3:**
>
> We appreciate the concern for a fair comparison and would like to clarify our experimental protocol, which we believe is fair and rigorous.
>
> - **For the Non-Interactive Task (Table 2):** The comparison is entirely fair. As stated in **Section 4.1 (lines 367-369)**, our model was trained and evaluated **exclusively on the public ViViD dataset** for this benchmark. No extra data from VVT-Interact was used. The performance gains here are a direct result of our model's architectural efficiency and advanced training strategies (detailed in Appendix A.6).
> - **For the Interactive Task (Table 1):** This is our newly proposed task, and VVT-Interact is the first dataset for it. Retraining baseline methods is problematic due to the lack of official open-source training code. Therefore, we directly tested existing methods on VVT-Interact to assess their **generalization capability** to this new complex task. This approach fairly demonstrates that prior methods are fundamentally unable to handle human-garment interactions.
>
> **W4:**
>
> Thank you for this practical suggestion. We agree that providing concrete efficiency metrics is crucial. While our model is parameter-efficient (2B), this efficiency also translates to tangible benefits in inference speed and memory usage. We have benchmarked iTryOn's performance and will add the following comparison to the paper:
> | Method | Parameters | Inference Time (per video) | VRAM Usage|
> | :--- | :---: | :---: | :---: |
> | MagicTryOn | 14B | 461 sec | 61.9 GB |
> | **iTryOn (ours)** | **2B** | **187 sec** | **38.4 GB** |

---

> > ### Author Response · Authors · 2025-11-24
> >
> > **W5:**
> >
> > We fully agree with the reviewer that quantifying physical plausibility in interactions is a significant challenge in this nascent field. While standard metrics (VFID, SSIM) provide a necessary baseline that generally correlates with visual quality, we acknowledge they are insufficient for measuring interaction correctness. To address this, we have taken two specific steps:
> >
> > 1. **Human Evaluation:** To provide the most reliable assessment of interaction fidelity, we conducted a user study on the 132 videos in our test set. Human evaluators compared iTryOn against the strongest competitor, MagicTryOn. The results were decisive: **iTryOn won in 81.9% cases, tied in 9.8%, and lost in only 8.3%**. This confirms that human observers perceive our interactions as significantly more plausible.
> > 2. **Exploration of VLM-based Metrics:** We actively explored a novel automated metric, "Interaction Precision." This method uses a VLM to frame-by-frame match the interaction occurrence in the generated video against the ground truth (calculating a score X/N, where N is the number of ground truth interaction frames and X is the successfully detected generated interaction frames). While iTryOn scored higher than competitors, we found that current VLMs are inconsistent when differences are subtle, making the metric unstable for fine-grained evaluation.
> >
> > We will include the detailed Human Evaluation results in the main text and add our exploration of the VLM-based metric to the **Appendix** to provide insights and a baseline for future research into specialized interaction metrics.
> >
> > **W6:**
> >
> > We appreciate the reviewer pointing this out. The current distribution is not accidental. It faithfully reflects the **natural data distribution of the live-streaming e-commerce domain**, which is the primary setting for this task. Actions like "Adjusting the collar" and "hem" are overwhelmingly the most frequent real-world interactions in this context.
> >
> > However, we agree that a balanced dataset is valuable for evaluating broader generalization capabilities. To address this:
> >
> > 1. We will explicitly state in the paper that the current distribution is domain-specific.
> > 2. For the final public release of the VVT-Interact dataset, we are currently augmenting the data to curate a **balanced subset** that upsamples rare interactions. This will ensure the community has access to both a representative "in-the-wild" distribution and a balanced distribution for robust algorithmic benchmarking.

---

### Official Review · Reviewer_axoj · 2025-11-01

**Soundness:** 2
**Presentation:** 3
**Contribution:** 3
**Rating:** 4
**Confidence:** 5

**Summary:**

This paper introduces a video tryon method where the human interacts with the garments noticeably. It is done by (1) modeling the hands explicity with 3D hand mesh model, and (2) a action-position-encoding to bias the generation on the motion rather than un-action words. They also introduce a dataset specifically for tryon with hands-cloth interaction. They showed improved results over prior state of art for video tryons.

**Strengths:**

1. Supporting human interaction with the garments is a important yet natural task for virtual tryon. The VVT-interact dataset is an important next step towards this goal.
2. The action aware semantic guidance is necessary to avoid the model from being biased by non-action words in the prompt.

**Weaknesses:**

1. The proposed method requires an existing video to run tryon. Depending on the motion of this video, there could be incompatiblity with the tryon garments,  say roll up sleeve motion when the garment is a short-sleeve, or unzip the jacket when the garment is a t-shirt. Ideally, we should be able to use text prompt to control the motion of the user in the source video such that we can select the garment that is compatible with the prompt.

2. The method uses explicit 3D hand mesh to condition try video generation model. This estimated hand model is often misaligned with the actual image. This is problematic because the video has to generate the hand that matches the rgb hand pixels of the exiting source video.

3. It was unclear why the existing clothing is not "delcoth", leading to bleeding problem seen in Fig. 3

**Questions:**

1. Line 200-201: looks like the action caption is determined by per-frame by a VLM. It is hard to determine action or motion with only 1 frame.
2. Vivid is a dataset with limited to no hands and cloth interaction. Why does the model outperform existing method on this data, as shown in Tab 2, especially with the smallest model capacity (2B)?

---

> ### Author Response · Authors · 2025-11-24
> **Response to Reviewer axoj**
>
> **Regarding Weaknesses:**
>
> **W1:**
>
> We'd like to clarify the scope and definition of the Video Virtual Try-On (VVT) task.
>
> - **Task Definition:** As established in the VVT field, the primary goal is to **replace a garment within a given source video while preserving the subject's original identity, motion, and the background**. This is fundamentally formulated as a video inpainting task conditioned on a reference garment image and the person's movements. This definition is consistent with prior seminal works in this domain, such as Fashion-VDM (Karras et al., 2024) and CatV2TON (Chong et al., 2025). The task you described is a fascinating but distinct research problem that falls outside the conventional definition of VVT.
> - **Handling Incompatibility:** We have investigated the model's behavior in cases of motion-garment incompatibility (e.g., applying an "unzipping a jacket" motion to a T-shirt). In such scenarios, iTryOn gracefully degrades to a non-interactive VVT result. The model preserves the original human motion (the hand gesture for unzipping) while realistically rendering the new garment (the T-shirt). The outcome is akin to a "pantomime" of the action, which is an acceptable and logical result within the task's constraints.
>
> **W2:**
>
> We address this concern from two main perspectives. **Firstly,** we took meticulous steps to ensure the high quality of the 3D hand prior itself. During the creation of our VVT-Interact dataset, we employed a VLM to filter out video clips with significant hand occlusion. As detailed in **Appendix A.4 (3D HAND PRIOR ANNOTATION)**, a subsequent manual inspection confirmed that the estimated 3D hand meshes achieve a high accuracy rate of over 95%. **Secondly,** our framework is architecturally designed for robustness against potential minor inaccuracies. It utilizes a synergistic, dual-guidance mechanism: the 3D hand mesh provides fine-grained detail, while the **DWpose features** simultaneously offer a reliable, foundational representation of the hand's position and overall body pose. This ensures that even if the 3D prior is slightly imperfect, the model receives sufficient and complementary information to generate a plausible and coherent result.
>
> **W3:**
>
> We apologize for the confusion caused by the figure's labeling and thank you for pointing this out. The model's input is indeed a **clothing-agnostic representation** where the original garment is masked out, as illustrated in our main framework diagram (Figure 2). The issue arose from the captioning in Figure 3. The column labeled "Input Video" was intended to show the **original source video** for reference, to make the comparison clearer for the reader. A more accurate label would have been "Source Video". We acknowledge this was a poor choice of words that led to ambiguity. We will revise the label in Figure 3 to **"Source Video"** in the final version of the paper to prevent any future misinterpretation.
>
> ---
>
> **Regarding Questions:**
>
> **Q1:**
> We would like to clarify that the action caption is **not** determined on a per-frame basis. The process is similar to that of generating the global caption: the **video clip is fed into the VLM in a single pass** to obtain a categorical action label. This allows the VLM to leverage the full temporal context of the video to make an accurate determination. A detailed explanation of this annotation process is provided in **Appendix A.3.1 (VLM-BASED ANNOTATION FOR SEMANTIC GUIDANCE)**.
>
> **Q2:**
>
> We anticipated this question and have provided a detailed analysis in **Appendix A.6 (ANALYSIS OF STATE-OF-THE-ART PERFORMANCE ON NON-INTERACTIVE VVT)**.
>
> The superior performance stems from two key factors, independent of our novel interaction-specific modules: (1) **A Superior Foundational Backbone:** Our choice of **Wan2.1-VACE** as the base model is strategic. Unlike some other generative backbones, Wan2.1-VACE is a versatile controllable video synthesis model pre-trained on tasks like reference-guided editing. This pre-training aligns perfectly with the VVT task, which is essentially a specialized form of video inpainting. Consequently, our framework inherits a powerful prior for maintaining garment fidelity and temporal coherence even on non-interactive tasks. (2) **Advanced Training and Inference Strategies:** We employ several powerful general-purpose techniques that significantly boost performance. As detailed in Appendix A.6, these include a specific **loss weighting** scheme during training to accelerate convergence and the use of **Interval Guidance** during inference to improve sample quality.
>
> In summary, our state-of-the-art results on the ViViD benchmark are attributable to the synergistic combination of a highly suitable foundational model and advanced general-purpose training strategies. This confirms that iTryOn is not only a novel solution for interactive VVT but also a highly efficient and powerful framework for the traditional VVT task.

---

### Author Response · Authors · 2025-12-04
**Summary of Paper Reviews and Rebuttal**

Dear Area Chair,

We thank you for handling our submission. Below, we briefly summarize the key strengths of our work and address the main concerns; detailed evidence and experiments are provided in the rebuttal.

Principal Strengths

1. New task and first dedicated dataset.

We introduce Interactive Video Virtual Try‑On (Interactive VVT), which explicitly requires modeling physically plausible human–garment interactions in video (e.g., pulling, stretching, adjusting), extending conventional VVT that only replaces garments. To support this task, we curate VVT‑Interact, the first dataset with explicit interaction categories and frame‑level temporal/text annotations. Existing VVT methods, evaluated zero‑shot on VVT‑Interact, largely fail to reproduce the required interactions, underscoring the necessity of our formulation.

2. Simple yet essential plug‑and‑play guidance.

Rather than altering the diffusion transformer backbone, we add lightweight guidance modules that enable capabilities the base model lacks:
(1) Spatial guidance (3D hand mesh + DWpose) focuses on contact regions while remaining garment‑agnostic and avoiding leakage of source garment geometry.
(2) Semantic/temporal guidance (global/action captions + A‑RoPE) aligns action text to the correct temporal segments, affecting only interaction frames.
(3) Action‑aware loss strengthens supervision on interactive frames.
Ablations show that a fine‑tuned Wan2.1‑VACE alone degenerates to non‑interactive try‑on; only with our guidance can it synthesize coherent human–garment interactions.

3. State‑of‑the‑art performance with high efficiency.

Our 2B‑parameter model achieves strong results on both interactive and standard VVT:
(1) On VVT‑Interact, baselines mostly yield non‑interactive outputs; a human study shows iTryOn is preferred to MagicTryOn in 81.9% of cases.
(2) On ViViD, trained strictly on public ViViD data, we obtain state‑of‑the‑art quantitative performance, surpassing 14B‑parameter baselines.
(3) On an NVIDIA A100, iTryOn is approximately 2.5× faster and uses notably less VRAM than MagicTryOn, improving practicality.

Brief Responses to Core Concerns

1. Novelty vs. backbone and additional data.

On ViViD, we use no extra data; gains come from backbone choice and training strategy. On VVT‑Interact, ablations show that Wan2.1‑VACE, even when fine‑tuned, fails to model interactions without our spatial and semantic guidance plus the action‑aware loss, indicating that our guidance—not just a stronger backbone or more data—drives the improvement.

2. 3D hand priors.

3D hand meshes are filtered and manually checked (>95% accuracy) and combined with DWpose for robust spatial guidance. Hands are the primary manipulators, and the 3D hand prior is garment‑agnostic, avoiding depth‑based geometric leakage.

3. Captions, alignment, and interaction types.

All captions and temporal segments are automatically extracted by a VLM from the input video (no user input). A‑RoPE temporally localizes action captions to interactive segments. Our concise interaction taxonomy is tailored to e‑commerce; the “Other interactions” category already covers unseen behaviors in practice, and we plan to expand it in future versions of VVT‑Interact.

4. Dataset, metrics, and ethics.

The class distribution reflects real live‑commerce behavior; we will also release a balanced subset. Beyond standard metrics, we report a human study and an exploratory VLM‑based interaction metric as first steps toward specialized evaluation. Data collection complies with platform terms, and we will mask faces in the released dataset to protect identities.

In summary, our main contribution is not a new backbone, but a compact, principled guidance framework that (i) enables a new, more challenging Interactive VVT task and (ii) achieves state‑of‑the‑art performance and efficiency on both interactive and non‑interactive video virtual try‑on.

Sincerely,

The Authors

---

### Meta-Review · Area_Chair_P4VT · 2026-01-14

**Summary:**

This paper introduces an interactive video virtual try-on framework that incorporates two key components: (1) explicit modeling of hands using a 3D hand mesh to guide interaction-aware generation, and (2) an action–position encoding mechanism that biases the generation toward motion-related cues rather than non-action textual tokens. In addition, the authors present a new dataset tailored for video virtual try-on scenarios involving hand–garment interactions.

**Reviewer Concerns:**

The reviewers raised several concerns regarding the proposed interactive video virtual try-on framework. Conceptually, the method relies heavily on a given source video and predefined interaction prompts, raising questions about motion–garment compatibility, the practicality and clarity of obtaining action prompts, and the generalizability of defining interactions primarily through hand motion and captions. Technically, the work was criticized for limited architectural novelty beyond existing diffusion-based try-on models, insufficient ablation and hyperparameter analysis, potentially unfair comparisons due to extra training data, and a lack of computational efficiency analysis. From an evaluation perspective, reviewers noted dataset imbalance, small and limited test benchmarks, absence of interaction-specific physical plausibility metrics, and visible artifacts or fidelity issues in results, along with unclear handling of implausible interactions and limited diversity in body shapes and interaction types.

**Reviewer Scores:**

The reviewers acknowledged several strengths of this work, notably that the proposed framework enables more realistic garment–person interactions and improved try-on visual effects compared to previous methods. At the same time, they raised several major concerns, which are also reflected in the authors’ rebuttal, particularly regarding the aspects like limited technical novelty and the reliance on additional data. Overall, while the reviewers’ ratings are somewhat mixed, the general sentiment tends to lean toward a negative assessment.

---

### Decision · Program_Chairs · 2026-01-26

Reject